# FactoryNet: A Large-Scale Dataset toward Industrial Time-Series Foundation Models

Karim Othman [* 1 2]   Jonas Petersen [* 3 2]   Matei C. Ignuta-Ciuncanu [4 5]   Alessandro Lombardi [2]
Camilla Mazzoleni [2]   Federico Martelli [3 2]   Riccardo Maggioni [2]   Philipp Christian Petersen [6]

🤗 Dataset: Forgis/FactoryNet
Code: github.com/Forgis-Labs/FactoryNet

## Abstract

We introduce the first universal pretraining corpus for industrial time-series data: **FactoryNet**. 51M datapoints across 23k end-to-end task executions (13.3k real, 9.8k synthetic) on six embodiments, unified by a shared schema that enables zero-shot cross-embodiment transfer and highly parameter-efficient anomaly detection. We introduce a novel schema: Setpoint, Effort, Feedback, Context (S-E-F-C) underlying the whole pipeline that maps any actuated system into a common representational frame. The corpus spans 27 annotated anomaly types alongside healthy baselines and counterfactual pairs across robotic manipulation and machining domains. Cross-embodiment transfer experiments yield positive results: under bias-aware metrics our model demonstrates fair cross-embodiment transfer capabilities on the evaluated source-target pair, while 24 schema-aligned signals achieve competitive anomaly detection performance compared to high-dimensional baselines. We release FactoryNet as a growing, multi-embodiment dataset to drive progress toward industrial foundation models.

## 1. Introduction

The manufacturing sector accounts for approximately 15% of global GDP, relying heavily on the continuous operation of complex actuated machinery (World Bank, 2026). While predictive maintenance and process optimization present significant opportunities for machine learning, industrial AI remains largely confined to single-machine, bespoke deployments. Foundation models have transformed vision and language by pretraining on large, structurally coherent corpora, yet no analogous substrate exists for industrial time-series. The gap is not merely volume: existing anomaly detection and forecasting datasets (Brockmann et al., 2024; Leporowski et al., 2021) record sensor outcomes without separating *commanded intent* from *measured response*. For actuated systems, learning transferable dynamics requires observing the full control loop from target trajectory through actuation effort to the resulting physical state yet no open dataset provides this decomposition across multiple embodiments.

We introduce **FactoryNet**, a large-scale dataset that enforces a control-theoretic decomposition to serve as a pretraining substrate for industrial foundation models. The corpus spans 23k end-to-end task executions (9,114 real laboratory episodes on UR3 and KUKA KR10, 4,185 standardized open-source episodes, and 9,799 synthetic episodes from Isaac Sim) on six embodiments. Every signal is mapped into the Setpoint-Effort-Feedback-Context (S-E-F-C) schema: a machine-agnostic signal taxonomy grounded in IEC 81346 functional classification that separates intent from outcome across arbitrary actuated systems. The dataset includes 27 annotated anomaly types alongside healthy baselines and counterfactual pairs. We validate that structural alignment provides a useful inductive bias: 24 schema-aligned signals match 130 unstructured ones at comparable AUROC (83.2% [81.5, 86.1]), while a TCN-Transformer forward-dynamics model achieves an average of 156.7 prediction steps (78.4% of the 200-step horizon at 100 Hz), substantially outperforming all unstructured baselines (TCN: 33.5 steps; Flat MLP: 20.8 steps; Linear: 10.7 steps). Under bias-aware metrics the TCN-Transformer also outperforms all baselines in zero-shot cross-embodiment transfer.

---

[*]Equal contribution  [1]Cairo University [2]Forgis [3]ETH Zurich [4]Imperial College London [5]UC Berkeley [6]University of Vienna. Correspondence to: Jonas Petersen <jep79@cantab.ac.uk>.

*Proceedings of the 2nd Workshop on Foundation Models for Structured Data at the 43rd International Conference on Machine Learning (FMSD@ICML 2026)*, Seoul, South Korea. 2026. Copyright 2026 by the author(s).

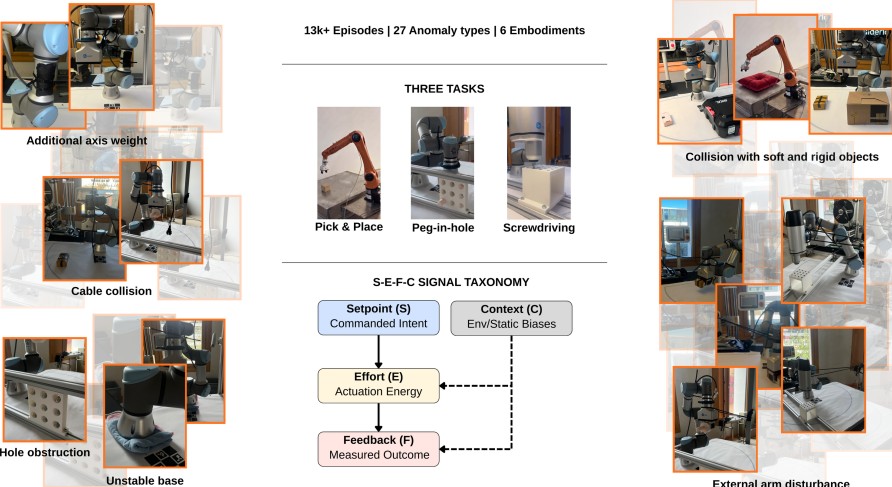

*Figure 1.* **FactoryNet overview.** A large-scale, multi-embodiment industrial time-series corpus spanning 13 k+ real episodes, 27 anomaly types, and 3 manipulation tasks. Every signal is mapped into the Setpoint-Effort-Feedback-Context (S-E-F-C) taxonomy, a control-theoretic decomposition that enables cross-embodiment learning. Representative fault types are shown alongside the corresponding laboratory setups.

## 2. Related Work

### 2.1. Industrial Fault-Detection Datasets

The availability of open-source data in the manufacturing domain lags significantly behind other modalities. Most existing datasets focus on single-machine, run-to-failure scenarios or specific rotating machinery components. Canonical examples include the NASA C-MAPSS turbofan degradation benchmark (Saxena et al., 2008) and recent robotic datasets: voraus-AD (Brockmann et al., 2024) provides 2,122 episodes of industrial robot recordings and AUR-SAD (Leporowski et al., 2021) offers 2,045 episodes. However, these datasets are bounded in scale and lack an explicit control-loop structure mapping commands to outcomes. By unifying these sources alongside novel laboratory data under a standard control-theoretic decomposition, FactoryNet provides the largest open-source, fault-injected industrial robot dataset to date (Table 4).

### 2.2. Foundation Models for Time-Series and Robotics

Current foundation model research diverges into two distinct tracks. In robotics, efforts such as Open X-Embodiment (O'Neill et al., 2024) and DROID (Khazatsky et al., 2024) pool data across diverse kinematics to train generalizable, cross-embodiment behavioral policies. Simultaneously, in the structured time-series domain, models such as Chronos (Ansari et al., 2024), TimesFM (Das et al., 2024), Moirai (Woo et al., 2024), MOMENT (Goswami et al., 2024), Timer (Liu et al., 2024), Lag-Llama (Rasul et al., 2023), and TabPFN-TS (Hoo et al., 2025) leverage pretraining corpora to yield powerful zero-shot forecasters. However, because these TS-FMs are predominantly

trained on web-scraped, financial, or ecological data, we hypothesize that they lack grounding in physical actuation and may struggle to disentangle static payload biases from actual dynamic behavior in industrial settings. FactoryNet is designed as a bridge between these two tracks: an industrial-scale, control-theoretically structured corpus intended to train and evaluate TS-FMs on complex, cross-embodiment mechanical dynamics.

## 3. The FactoryNet Dataset

FactoryNet v1.0 is the largest open-source industrial-robot time-series dataset containing labelled anomalies and organized around a control-theoretic schema. It comprises three pillars: real-world laboratory recordings, standardized open-source adaptations, and a synthetic generation pipeline. Because industrial data is sampled at high frequencies (100 Hz) across many joints and sensors, 23k episodes yield a high-density, continuous corpus of the kind required to learn complex physical dynamics.

### 3.1. Dataset Composition

As detailed in Table 1, the corpus encompasses multiple data streams. The laboratory track includes extensive, high-frequency (100 Hz) recordings from UR3 and KUKA KR10 industrial robotic arms executing three complex tasks: Pick & Place, Screwdriving, and Peg-in-Hole. We ingested and restructured existing high-quality open-source datasets (voraus-AD, AURSAD, and UMich CNC (Sun, 2018)) into our unified schema. The dataset is structured using a hierarchical taxonomy grounded in the IEC 81346 standard for industrial systems, allowing the schema to naturally inte-

grate diverse modalities such as CNC milling centres and rotating machinery. A parallel synthetic track generated via NVIDIA Isaac Sim provides procedurally scaled data for model pretraining.

*Table 1.* FactoryNet Composition. Datapoint counts are approximate; each time-step record contains ∼20–40 scalar channels.

| Source | Machine | Tasks | Faults | Episodes | Datapoints |
|---|---|---|---|---|---|
| **Our Lab (Real)** | UR3 | P&P, Screw, Peg | Yes | 7,141 | 18M |
| **Our Lab (Real)** | KUKA KR10 | P&P | Yes | 1,973 | 4M |
| **Open (Real)** | voraus-AD (Yu-Cobot) | P&P | Yes | 2,122 | 16M |
| **Open (Real)** | AURSAD (UR3e) | Screw | Yes | 2,045 | 3M |
| **Open (Real)** | UMich CNC | Machining | Yes | 18 | 18K |
| **Synthetic** | Isaac Sim (UR5) | P&P | No | 9,799 | 10M |

### 3.2. The S-E-F-C Signal Taxonomy: A Unified Vocabulary

The defining feature of FactoryNet is its control-theoretic structure. Standard datasets record sensor readings without separating commanded intent from measured outcome, conflating controller targets with physical responses and suffering from vendor-specific naming conventions. We map over 300 unique, heterogeneous data columns into a standardized tensor format across four distinct roles: **Setpoint (S)** (commanded intent, e.g., target joint positions and velocities), **Effort (E)** (actuation energy expended by the controller, e.g., motor current and torque), **Feedback (F)** (measured physical outcome, e.g., actual joint positions), and **Context (C)** (environmental or static variables, e.g., payload mass, TCP configuration).

This taxonomy allows a single dataloader to work across UR3e, KUKA KR10, and CNC machinery: a 6-DOF rotational arm and a 3-axis linear gantry expose the same four roles, only with different axis counts and units. By enforcing this taxonomy, S-E-F-C acts as a unified vocabulary for cross-embodiment models, providing a principled inductive bias for learning the difference between expected dynamics and external disturbances. Signal availability varies by embodiment: KUKA KR10 (KSS 8.3) does not expose joint velocities, commanded TCP pose, or TCP force/torque via its RSI interface; these channels are marked absent in the S-E-F-C mapping (Appendix D).

### 3.3. Synthetic Pipeline and Sim-to-Real

Relying solely on real-world data bottlenecks the scaling required for foundation model pretraining. FactoryNet includes a dedicated synthetic pipeline built on NVIDIA Isaac Sim. Utilizing procedural generation and domain randomization (varying mass, friction, and controller gains), we provide 9,799 simulated Pick & Place episodes with programmatically aligned S-E-F-C metadata. For each synthetic episode a counterfactual healthy twin is generated under identical conditions, enabling controlled analysis of

fault-induced signal deviations. To quantify the sim-to-real gap we replay real `target_joint_*` waypoints in Isaac Sim and pair simulated with real episodes by ID; phase-wise gap metrics over 1,155 paired episodes are summarized in Table 5 (full domain-randomization ranges and protocol in Appendix H).

### 3.4. Faults and Anomalies

To support anomaly detection and robust control research, of the 9,114 lab episodes, approximately 40% are healthy and 60% contain injected faults across 27 anomaly types spanning three tasks: **Pick & Place** (payload weight variations, misgrips, environmental collisions), **Screwdriving** (stripped threads, misaligned insertions, torque-limit violations), and **Peg-in-Hole** (jamming, altered friction coefficients, orientation offsets).

### 3.5. Data Accessibility and Licensing

Novel laboratory and synthetic data are released under the MIT license; adapted open-source subsets retain their original licenses (CC-BY 4.0 or equivalent). The repository provides S-E-F-C Parquet files, metadata, and framework-native dataloaders at `https://huggingface.co/datasets/Forgis/FactoryNet`.

## 4. Dataset Utility & Validation

To demonstrate that FactoryNet provides a viable substrate for both single-machine modelling and foundation model pretraining, we evaluate the dataset across standard industrial baselines and establish the open challenge of cross-embodiment transfer.

**Evaluation Protocol.** All experiments use a fixed train/validation/test split. For voraus-AD anomaly detection we follow the official protocol of Brockmann et al. (2024): training on 948 healthy episodes only, testing on the 1174-episode labelled set (419 healthy + 755 anomalous). Confidence intervals for the FactoryNet MLP are 95% bootstrap CIs computed over 1,000 resamplings of episode-level anomaly scores; CIs for unstructured baselines are reported as standard deviation across fault categories as published in Brockmann et al. (2024). Cross-machine transfer uses zero-shot evaluation with no target-domain fine-tuning.

### 4.1. Single-Machine Baselines: Validating the S-E-F-C Schema

**Anomaly Detection.** We use the voraus-AD subset to evaluate whether the S-E-F-C schema enables competitive anomaly detection with far fewer features. An MLP predicts the 6 motor-torque Effort signals from 18 Setpoint signals (position, velocity, acceleration); its per-episode

mean absolute error is the anomaly score. Operating on only 24 S-E-F-C signals, it reaches a mean AUROC of **83.2% [81.5, 86.1]** (95% bootstrap CI). As shown in Table 2, this outperforms weaker unstructured baselines that use all 130 signals (1-NN, GANF, PCA), while remaining below the stronger full-signal methods (CAE, LSTM-VAE, MVT-Flow). These results demonstrate that the S-E-F-C schema provides a useful signal-selection prior, achieving competitive performance with ~5× fewer features.

*Table 2.* Mean AUROC on voraus-AD anomaly detection. Unstructured baselines reproduced from Brockmann et al. (2024) (all use 130 signals). Our methods use only 24 S-E-F-C signals. [†]95% bootstrap CI over episodes. [‡]Std across 12 fault categories (from source paper).

| Method | Signals | Mean AUROC |
|---|---|---|
| 1-NN | 130 | 77.5 |
| GANF | 130 | $79.9 \pm 12.7^{\ddagger}$ |
| PCA | 130 | 80.0 |
| **S-E-F-C MLP (Ours)** | **24** | **83.2 [81.5, 86.1]**[†] |
| CAE | 130 | $85.2 \pm 9.2^{\ddagger}$ |
| LSTM-VAE | 130 | $86.7 \pm 10.1^{\ddagger}$ |
| MVT-Flow | 130 | $93.6 \pm 5.7^{\ddagger}$ |

**Multi-Step Forecasting.** To demonstrate that the dataset supports high-fidelity dynamics modelling, we evaluate an autoregressive TCN-Transformer on the voraus-AD (Yu-Cobot) Pick & Place data. The TCN-Transformer comprises a 3-layer TCN (kernel size 3) and a 2-layer Transformer (4 heads, hidden dimension 64, containing approximately 105,000 parameters). It is trained using a context window of 10 steps to predict 1 step ahead. Models are trained on 1,093 normal episodes and validated on 137 held-out normal episodes (sampled at 100 Hz).

The model predicts joint acceleration, which is integrated (Euler, $\Delta t = 0.01$ s) to derive position and velocity. As shown in Table 6 and Figure 3, the TCN-Transformer holds a strict 0.01 rad per-joint position error for an average of **156.7 steps** (78.4% of the 200-step horizon at 100 Hz), versus 33.5 (TCN), 20.8 (Flat MLP), and 10.7 (Linear); at 200 steps its MSE ($0.11 \times 10^{-4}$ rad$^2$) is more than four orders of magnitude below the next best baseline.

### 4.2. Cross-Embodiment Transfer: An Open Challenge

The S-E-F-C schema enables structured zero-shot transfer across machine types. We formally define the *mean-centered MAE* metric, which isolates dynamic forces from static payload biases:

$$\text{MC-MAE} = \frac{1}{EJT}\sum_{e=1}^{E}\sum_{j=1}^{J}\sum_{t=1}^{T}\left|\left(y_{e,j,t}-\bar{y}_{e,j}\right)-\left(\hat{y}_{e,j,t}-\bar{\hat{y}}_{e,j}\right)\right|,$$
(1)

where $\bar{y}_{e,j}$ and $\bar{\hat{y}}_{e,j}$ denote the per-episode, per-joint mean of the ground truth and prediction, respectively, $E$ is the number of episodes, $J = 6$ joints, and $T$ is the episode length.

A TCN-Transformer trained solely on voraus-AD (Yu-Cobot) Pick & Place data (no target-domain fine-tuning) achieves a mean-centered MAE of $\mathbf{0.339 \pm 0.006}$ on 1,433 AURSAD (UR3e) Screwdriving episodes—outperforming every baseline, including the kinematic baseline ($0.373 \pm 0.005$) and all structureless learned models (Table 3). In raw survival-step terms the transfer failure is stark (6.2 steps zero-shot vs. 156.7 in-domain), but Eq. (1) reveals this collapse is driven by static joint-trajectory offsets between embodiments rather than an absence of shared dynamics. The dynamic shape *does* transfer; bias-aware evaluation is essential to measure it.

*Table 3.* Zero-shot cross-embodiment transfer (voraus-AD Yu-Cobot P&P → AURSAD UR3e Screwdriving, 1,433 episodes). MC-MAE (Eq. 1) removes per-joint trajectory bias; lower is better. 95% CI computed over episodes.

| Model | MC-MAE | 95% CI ($\pm$) |
|---|---|---|
| Linear | 0.928 | 0.023 |
| Flat MLP | 0.792 | 0.019 |
| TCN | 0.770 | 0.017 |
| Kinematic Baseline | 0.373 | 0.005 |
| **TCN-Transformer (Ours)** | **0.339** | **0.006** |

## 5. Conclusion & Future Work

FactoryNet provides the largest open-source, fault-injected time-series corpus for industrial robotics. By enforcing the S-E-F-C taxonomy and a hierarchical IEC standard, it equips industrial ML with the control-theoretic decomposition needed for structural dynamics modelling. Future work will expand the laboratory and synthetic datasets across new machine families and build a benchmark suite for cross-embodiment transfer and sim-to-real gap quantification.

**Limitations.** The S-E-F-C schema targets actuated systems with controller-exposed setpoints; extension to non-actuated machinery (e.g., passive bearings) is future work. The synthetic pipeline currently covers Pick & Place only, and cross-embodiment transfer is preliminary, with full benchmark protocols deferred to concurrent work. KUKA KR10 recordings lack joint velocities, commanded TCP pose, and TCP force/torque (RSI, KSS 8.3), yielding a sparser mapping than UR3e. Counts reflect the actively-growing corpus at submission time; all experiments are single-seed, and per-category results appear in Appendix E.

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

## A. Corpus Composition Overview

Figure 2 traces how FactoryNet's data sources map onto embodiments and target tasks.

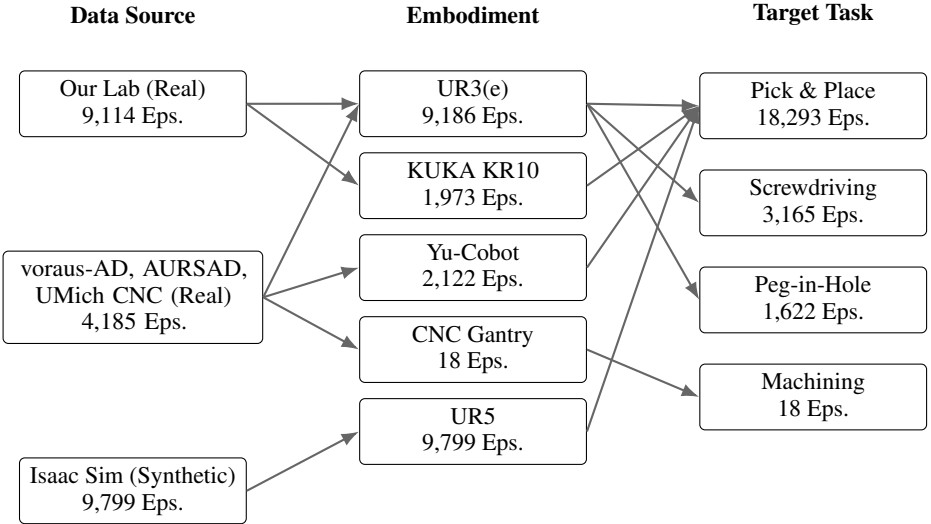

*Figure 2.* **The FactoryNet Dataset.** A structured overview of the corpus composition illustrating the mapping of 23k task executions. The dataset aggregates real-world laboratory recordings, standardized open-source subsets, and synthetic generations into a unified pretraining substrate covering four distinct actuation tasks.

## B. Additional Results

*Table 4.* Comparison of open-source industrial time-series datasets. Expanding beyond single-domain recordings, FactoryNet is the only multi-machine corpus requiring a strict control-loop structure decoupling intended Setpoint from applied Effort.

| Dataset | Year | Machine Type | Signals | Samples (Eps.) | Has Setpoint? | Has Effort? |
|---|---|---|---|---|---|---|
| CWRU (Loparo, 2000) | 2000 | Bearings | Vibration | ∼480 | No | No |
| PHM 2010 (PHM Society, 2010) | 2010 | CNC milling | Force, vibration | 315 | Partial | Yes |
| Paderborn (Lessmeier et al., 2016) | 2016 | Bearings | Vibration, current | ∼2,000 | No (RPM only) | Partial |
| MAFAULDA (Marins et al., 2016) | 2016 | Rotating machinery | Vibration, audio | 1,951 | No | No |
| XJTU-SY (Lei et al., 2019) | 2019 | Bearings | Vibration | 15 | No | No |
| AURSAD (Leporowski et al., 2021) | 2021 | UR3e robot | Joint signals | 2,045 | Yes | Yes |
| voraus-AD (Brockmann et al., 2024) | 2024 | Collaborative robot | 130 channels | 2,122 | Yes | Yes |
| **FactoryNet (Ours)** | **2026** | **Multi-machine** | **Multimodal** | **23k** | **Yes (Required)** | **Yes (Required)** |

*Table 5.* Batch sim-to-real gap over 1,155 paired episodes (pooled per-episode metrics).

| Metric | Mean | Median | P10 | P90 |
|---|---|---|---|---|
| Joint RMSE (deg) | 3.65 | 2.83 | 1.93 | 5.12 |
| TCP position RMSE (mm) | 13.16 | 13.26 | 8.87 | 16.92 |
| EE L2 RMS (mm) | 25.11 | 25.17 | 16.96 | 32.58 |
| W1 effort mean (A) | 0.83 | 0.81 | 0.77 | 0.91 |

## C. Datasheet for Datasets (Gebru et al., 2021)

Following the recommendations of Gebru et al. (2021), we provide a structured datasheet for FactoryNet v1.0 (Table 7).

*Table 6.* Multi-step forecasting on voraus-AD (Yu-Cobot) Pick & Place. MSE ($\times 10^{-4}$ rad$^2$) and MAE ($\times 10^{-2}$ rad) $\pm$ std over test episodes. Lower is better.

| Model | H = 50 | | H = 100 | | H = 200 | |
|---|---|---|---|---|---|---|
| | MSE | MAE | MSE | MAE | MSE | MAE |
| Linear | $28.17_{\pm 0.28}$ | $3.25_{\pm 0.02}$ | $541.22_{\pm 36.57}$ | $12.99_{\pm 0.27}$ | $6930.06_{\pm 67.51}$ | $51.31_{\pm 0.84}$ |
| Flat MLP | $1.69_{\pm 0.05}$ | $0.80_{\pm 0.02}$ | $67.02_{\pm 17.12}$ | $4.15_{\pm 0.46}$ | $1939.57_{\pm 160.39}$ | $24.43_{\pm 1.33}$ |
| TCN | $0.46_{\pm 0.03}$ | $0.46_{\pm 0.02}$ | $102.63_{\pm 30.85}$ | $4.68_{\pm 0.78}$ | $2940.49_{\pm 218.03}$ | $30.20_{\pm 1.73}$ |
| **TCN-Transformer** | $\mathbf{0.00_{\pm 0.00}}$ | $\mathbf{0.01_{\pm 0.00}}$ | $\mathbf{0.00_{\pm 0.00}}$ | $\mathbf{0.03_{\pm 0.00}}$ | $\mathbf{0.11_{\pm 0.01}}$ | $\mathbf{0.18_{\pm 0.01}}$ |

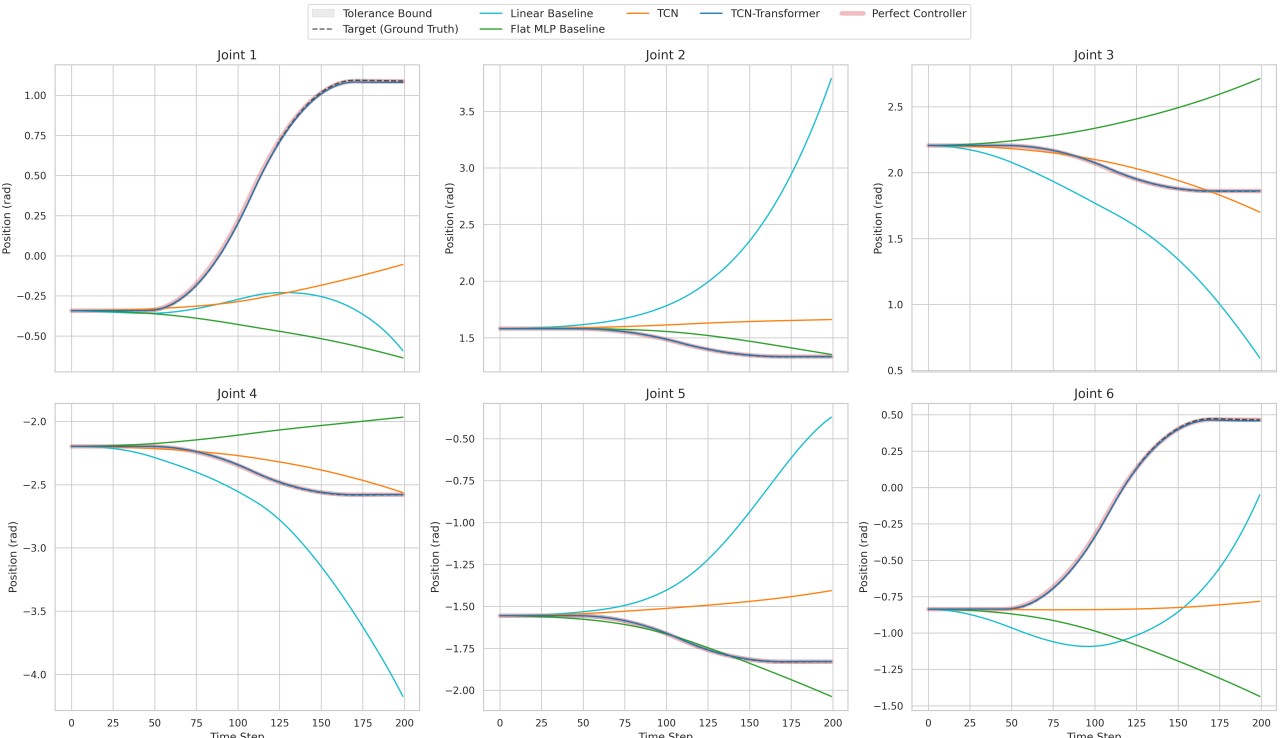

*Figure 3.* In-domain multi-step forecasting on the voraus-AD (Yu-Cobot) Pick & Place task. The TCN-Transformer maintains predictive accuracy within the 0.01 rad threshold for an average of 156.7 steps (78.4% of the 200-step horizon at 100 Hz), substantially outperforming all baselines.

## D. Signal-to-S-E-F-C Mapping

We provide one table per source mapping raw signal names to S-E-F-C roles. UR3, KUKA KR10 and Isaac Sim mappings are listed with raw signal names confirmed; CNC, AURSAD, and voraus-AD mappings are fully confirmed.

### B.1   UR3 Laboratory Source

Signals of UR3 can be found at Table  8

### B.2   KUKA KR10 Laboratory Source

Signals of KUKA KR10 can be found at Table  9

*Table 7.* Datasheet for FactoryNet v1.0.

| Question | Answer |
|---|---|
| *Motivation* | |
| Purpose | Pretraining substrate for industrial time-series foundation models, anomaly detection benchmarking, and cross-embodiment transfer research. |
| Creators | Karim Othman (Forgis), Jonas Petersen (ETH Zurich / Forgis), Matei C. Ignuta-Ciuncanu (Imperial College / UC Berkeley), Alessandro Lombardi (Forgis), Camilla Mazzoleni (Forgis), Federico Martelli (ETH Zurich / Forgis), Riccardo Maggioni (Forgis), Philipp Christian Petersen (University of Vienna). |
| Funding | Fully funded by Forgis AG. |
| *Composition* | |
| Instances | Episodes of robotic task execution, each containing multi-channel time-series at 100 Hz with S-E-F-C role annotations. |
| Count | 23,098 episodes total: 7,141 real lab (UR3), 1,973 real lab (KUKA KR10), 4,185 open-source (voraus-AD 2,122; AURSAD (UR3e) 2,045; UMich CNC 18), 9,799 synthetic (Isaac Sim UR5). |
| Sampling | Lab data: programmatic task execution with configurable fault injection (40% healthy, 60% anomalous). Open-source: full datasets ingested. Synthetic: procedural generation with domain randomization. |
| Missing data | Sensor dropouts <0.1% of time steps; handled via linear interpolation. No episodes removed due to missing data. |
| Confidentiality | No PII. Industrial parameters only (joint angles, torques, velocities). |
| *Collection Process* | |
| Acquisition | Lab episodes recorded by the authors via robot controller APIs (RTDE for UR3, RSI for KUKA KR10) at 100 Hz. Open-source subsets downloaded and re-processed into S-E-F-C format via adapter scripts. |
| Mechanisms | Automated scripts executing task programs with fault injection at configurable rates. 27 anomaly types injected programmatically (payload variation, misgrips, collisions, stripped threads, jamming, friction changes, etc.). |
| Who collected | Yanis Merzouki, Karim Othman, Balazs Gunther. |
| Time frame | March 1 – April 20, 2026. |
| Ethical review | N/A — no human subjects. |
| *Preprocessing, Cleaning, and Labeling* | |
| Preprocessing | Resampling to 100 Hz, NaN interpolation, S-E-F-C column mapping per source via adapter scripts. No per-channel normalisation applied (raw physical units preserved). |
| Raw data | Raw recordings available on request. Distributed format: S-E-F-C Parquet files with metadata. |
| Labeling | Programmatic fault injection with known ground truth; no manual annotation required for lab data. Open-source subsets retain original labels, mapped to unified anomaly taxonomy. |
| *Uses* | |
| Prior uses | Anomaly detection baselines (this paper), cross-embodiment transfer experiments (this paper). |
| Other uses | Forecasting, remaining useful life estimation, control policy learning, sim-to-real benchmarking, counterfactual analysis. |
| Misuse potential | Dataset reflects specific robot models and lab conditions; generalisation claims to unseen embodiments or industrial deployments should be scoped accordingly. |
| *Distribution* | |
| Distribution | Available on HuggingFace at https://huggingface.co/datasets/Forgis/FactoryNet. |
| License | Novel lab and synthetic data under MIT. Open-source subsets retain original licenses (CC-BY 4.0 or equivalent). |
| Restrictions | None. No export controls apply. |
| *Maintenance* | |
| Maintainer | Forgis AG. Issue tracker: GitHub repository. |
| Updates | Semantic versioning (v1.0 at submission). New embodiments and tasks increment minor version; schema changes increment major version. Prior versions remain available. |
| Retention | All versions permanently hosted on HuggingFace. |
| Errata | Reported and tracked via GitHub issues; corrections published as patch versions. |

## B.3 UMich CNC Source

Mapping of CNC columns (Table 10)

*Table 8.* S-E-F-C mapping for the UR3 laboratory source.

| Raw Signal Name | S-E-F-C Role | Unit | Notes |
| --- | --- | --- | --- |
| machine_id | Metadata | — | Episode metadata |
| ur3_robot_target_joint_0...5 | Setpoint | rad | 6 joints |
| ur3_robot_target_joint_vel_0...5 | Setpoint | rad/s | 6 joints |
| ur3_robot_joint_0...5 | Feedback | rad | Actual positions, 6 joints |
| ur3_robot_joint_vel_0...5 | Feedback | rad/s | Actual velocities, 6 joints |
| ur3_robot_joint_current_0...5 | Effort | A | 6 joints |
| ur3_robot_target_joint_current_0...5 | Setpoint | A | 6 joints |
| ur3_robot_joint_temp_0...5 | Context | °C | 6 joints |
| ur3_robot_joint_control_output_0...5 | Effort | — | 6 joints |
| ur3_robot_joint_mode_0...5 | Context | enum | 6 joints |
| ur3_robot_tcp_x/y/z | Feedback | m | Actual TCP position |
| ur3_robot_tcp_rx/ry/rz | Feedback | rad | Actual TCP orientation |
| ur3_robot_target_tcp_x/y/z | Setpoint | m | Target TCP position |
| ur3_robot_target_tcp_rx/ry/rz | Setpoint | rad | Target TCP orientation |
| ur3_robot_tcp_speed_x/y/z/rx/ry/rz | Feedback | m/s & rad/s | Actual TCP speed (6 axes) |
| ur3_robot_target_tcp_speed_x/y/z/rx/ry/rz | Setpoint | m/s & rad/s | Target TCP speed (6 axes) |
| ur3_robot_tcp_force_x/y/z | Effort | N | TCP force |
| ur3_robot_tcp_torque_x/y/z | Effort | Nm | TCP torque |
| ur3_robot_robot_mode | Context | enum | Enum |
| ur3_robot_safety_mode | Context | enum | Enum |
| ur3_robot_digital_inputs | Context | bitmask | Bitmask |
| ur3_robot_digital_outputs | Context | bitmask | Bitmask |
| ur3_robot_robot_status | Context | enum | |
| ur3_robot_safety_status | Context | enum | |
| ur3_robot_runtime_state | Context | enum | |
| ur3_robot_main_voltage | Context | V | V |
| ur3_robot_robot_voltage | Context | V | V |
| ur3_robot_robot_current | Context | A | A |
| ur3_robot_analog_input_0/1 | Context | V/A | |
| ur3_robot_analog_output_0/1 | Context | V/A | |
| ur3_robot_speed_scaling | Context | % | |
| ur3_robot_target_speed_fraction | Context | % | |
| ur3_robot_momentum | Context | kg·m/s | |
| realsense_camera_frame_count | Context | count | |
| realsense_camera_connected | Context | bool | |
| realsense_camera_streaming | Context | bool | |

*Table 9.* S-E-F-C mapping for the KUKA KR10 laboratory source. Note hardware limitations of KSS 8.3.

| Signal Group | Described Fields | S-E-F-C Role | Unit |
|---|---|---|---|
| Position (actual) | Actual joint positions (×6) | Feedback | degrees |
| Position (commanded) | Commanded joint positions (×6) | Setpoint | degrees |
| TCP pose (actual) | Actual TCP pose (×6) | Feedback | mm / degrees |
| Motor current | Motor current per axis (×6) | Effort | % of max |
| Motor torque | Motor torque per axis (×6) | Effort | Nm |
| Motor temperature | Motor temperature per axis (×6) | Context | Kelvin |
| Cartesian accel. | Accel. X, Y, Z, abs (×4) | Feedback | $m/s^2$ |
| Speed override | Speed override (scalar) | Context | % |
| Process state | State enum (FREE/ACTIVE/STOP/END) | Context | enum |
| Digital I/O | Digital inputs bitmask | Context | bitmask |
| Digital I/O | Digital outputs bitmask | Context | bitmask |

*Not available on KSS 8.3: joint velocities, commanded TCP pose, TCP force/torque, joint voltage, tool accelerometer, joint control output.*

*Table 10.* S-E-F-C mapping for the UMich CNC milling source. Axes: X1, Y1, Z1 (linear), S1 (spindle).

| Raw Signal Name | S-E-F-C Name | Role | Unit | Axis |
|---|---|---|---|---|
| X1_CommandPosition | setpoint_pos_0 | Setpoint | mm | X |
| X1_CommandVelocity | setpoint_vel_0 | Setpoint | mm/s | X |
| X1_CommandAcceleration | setpoint_acc_0 | Setpoint | $mm/s^2$ | X |
| X1_ActualPosition | feedback_pos_0 | Feedback | mm | X |
| X1_ActualVelocity | feedback_vel_0 | Feedback | mm/s | X |
| X1_ActualAcceleration | feedback_acc_0 | Feedback | $mm/s^2$ | X |
| X1_CurrentFeedback | feedback_current_0 | Feedback | A | X |
| X1_OutputCurrent | effort_current_0 | Effort | A | X |
| X1_OutputVoltage | effort_voltage_0 | Effort | V | X |
| X1_OutputPower | effort_power_0 | Effort | W | X |
| X1_DCBusVoltage | ctx_busvoltage_0 | Context | V | X |
| *(Pattern repeats for Y1 → axis 1, Z1 → axis 2)* | | | | |
| S1_CommandPosition | setpoint_pos_3 | Setpoint | deg | Spindle |
| S1_CommandVelocity | setpoint_vel_3 | Setpoint | rpm | Spindle |
| S1_CommandAcceleration | setpoint_acc_3 | Setpoint | rpm/s | Spindle |
| S1_ActualPosition | feedback_pos_3 | Feedback | deg | Spindle |
| S1_ActualVelocity | feedback_vel_3 | Feedback | rpm | Spindle |
| S1_ActualAcceleration | feedback_acc_3 | Feedback | rpm/s | Spindle |
| S1_CurrentFeedback | feedback_current_3 | Feedback | A | Spindle |
| S1_OutputCurrent | effort_current_3 | Effort | A | Spindle |
| S1_OutputVoltage | effort_voltage_3 | Effort | V | Spindle |
| S1_OutputPower | effort_power_3 | Effort | W | Spindle |
| S1_DCBusVoltage | ctx_busvoltage_3 | Context | V | Spindle |
| S1_SystemInertia | ctx_inertia_3 | Context | — | Spindle |
| M1_CURRENT_PROGRAM_NUMBER | ctx_program_number | Context | — | — |
| M1_sequence_number | ctx_sequence_number | Context | — | — |
| M1_CURRENT_FEEDRATE | ctx_feedrate | Context | mm/min | — |
| Machining_Process | ctx_process_phase | Context | enum | — |

### B.4  AURSAD Source (UR3e Screwdriving)

Per-joint signals follow a fixed pattern for joints $i \in \{0, \ldots, 5\}$; one representative joint (joint 0) is shown below. (Table 11)

*Table 11.* S-E-F-C mapping for the AURSAD source (representative joint 0 pattern; repeats for joints 1–5 with incremented index).

| Raw Signal Name | S-E-F-C Name | Role | Unit | Notes |
|---|---|---|---|---|
| target_q_0 | setpoint_pos_0 | Setpoint | rad | Joint 0 |
| target_qd_0 | setpoint_vel_0 | Setpoint | rad/s | |
| target_qdd_0 | setpoint_acc_0 | Setpoint | rad/s$^2$ | |
| target_current_0 | setpoint_current_0 | Setpoint | A | |
| target_moment_0 | setpoint_torque_0 | Setpoint | Nm | |
| actual_q_0 | feedback_pos_0 | Feedback | rad | |
| actual_qd_0 | feedback_vel_0 | Feedback | rad/s | |
| actual_current_0 | effort_current_0 | Effort | A | |
| actual_control_output_0 | effort_control_0 | Effort | — | |
| actual_joint_voltage_0 | effort_voltage_0 | Effort | V | |
| joint_temperatures_0 | ctx_temp_0 | Context | °C | |
| joint_mode_0 | ctx_joint_mode_0 | Context | enum | |
| target_TCP_pose_0 | setpoint_pos_cartesian_0 | Setpoint | m/rad | |
| target_TCP_speed_0 | setpoint_vel_cartesian_0 | Setpoint | m/s | |
| actual_TCP_pose_0 | feedback_pos_cartesian_0 | Feedback | m/rad | |
| actual_TCP_speed_0 | feedback_vel_cartesian_0 | Feedback | m/s | |
| actual_TCP_force_0 | effort_force_cartesian_0 | Effort | N | |
| actual_tool_accelerometer_0/1/2 | auxiliary_accel_tool_0/1/2 | — | m/s$^2$ | Tool IMU |
| output_double_register_24 | feedback_torque_tool | Feedback | Nm | Tool torque |
| output_double_register_25 | effort_torque_tool | Effort | Nm | |
| output_double_register_26 | setpoint_torque_tool | Setpoint | Nm | |
| output_double_register_27 | setpoint_torque_gradient_tool | Setpoint | Nm/s | |
| actual_main_voltage | ctx_main_voltage | Context | V | |
| actual_robot_voltage | ctx_robot_voltage | Context | V | |
| actual_robot_current | ctx_robot_current | Context | A | |
| speed_scaling | ctx_speed_scaling | Context | — | |
| target_speed_fraction | ctx_target_speed_fraction | Context | — | |
| actual_momentum | ctx_momentum | Context | kg·m/s | |
| robot_mode | ctx_robot_mode | Context | enum | |
| safety_mode | ctx_safety_mode | Context | enum | |
| runtime_state | ctx_runtime_state | Context | enum | |
| label | raw_label | — | — | 0=Normal, 1=Damaged screw, 2=Extra part, 3=Missing screw, 4=Damaged thread |

### B.5  voraus-AD Source (Yu-Cobot)

Per-joint signals follow the same index pattern for joints $i \in \{1, \ldots, 6\}$ (1-indexed in source); one representative joint is shown. (Table 12)

### B.6  Isaac Sim UR5 (Synthetic)

Mapping of Isaac Sim UR5 columns (Table 13)

## E. Per-Category Anomaly Detection Breakdown

Table 14 reports per-category AUROC on the voraus-AD subset for all seven methods compared in the main paper. The S-E-F-C MLP achieves the highest AUROC on mechanically distinctive faults (Miscommutation: 99.2, Additional Axis Weight: 95.8) where sustained Effort–Feedback divergence provides a strong discriminative signal, but struggles on transient or subtle gripping failures (Collision w/ Cables: 67.6, Losing Can: 71.8) where the anomaly window is brief and the single-step MLP lacks temporal modelling capacity.

†CI for S-E-F-C MLP is 95% bootstrap over episodes. ‡CI for other methods is std across fault categories (from Brockmann

*Table 12.* S-E-F-C mapping for the voraus-AD source (representative joint 1 pattern; repeats for joints 2–6).

| Raw Signal Name | S-E-F-C Name | Role | Unit | Notes |
|---|---|---|---|---|
| target_position_1 | setpoint_pos_0 | Setpoint | rad | Joint 1→idx 0 |
| target_velocity_1 | setpoint_vel_0 | Setpoint | rad/s | |
| target_acceleration_1 | setpoint_acc_0 | Setpoint | rad/s$^2$ | |
| target_torque_1 | setpoint_torque_0 | Setpoint | Nm | |
| joint_position_1 | feedback_pos_0 | Feedback | rad | |
| joint_velocity_1 | feedback_vel_0 | Feedback | rad/s | |
| motor_position_1 | feedback_motor_pos_0 | Feedback | rad | |
| motor_velocity_1 | feedback_motor_vel_0 | Feedback | rad/s | |
| torque_sensor_a_1 | feedback_torque_a_0 | Feedback | Nm | |
| torque_sensor_b_1 | feedback_torque_b_0 | Feedback | Nm | |
| motor_torque_1 | effort_motor_torque_0 | Effort | Nm | |
| motor_iq_1 | effort_current_iq_0 | Effort | A | |
| motor_id_1 | effort_current_id_0 | Effort | A | |
| power_motor_el_1 | effort_power_el_0 | Effort | W | |
| power_motor_mech_1 | effort_power_mech_0 | Effort | W | |
| power_load_mech_1 | effort_power_load_0 | Effort | W | |
| motor_voltage_1 | effort_voltage_0 | Effort | V | |
| computed_inertia_1 | ctx_inertia_0 | Context | — | |
| computed_torque_1 | ctx_computed_torque_0 | Context | Nm | |
| supply_voltage_1 | ctx_busvoltage_0 | Context | V | |
| brake_voltage_1 | ctx_brake_voltage_0 | Context | V | |
| robot_voltage | ctx_robot_voltage | Context | V | Global |
| robot_current | ctx_robot_current | Context | A | Global |
| io_current | ctx_io_current | Context | A | Global |
| system_current | ctx_system_current | Context | A | Global |
| anomaly | ctx_is_anomaly | — | bool | Label |
| category | ctx_anomaly_category | — | str | Fault type |
| setting | ctx_setting | — | str | |

et al. (2024)).

## F. Full Fault Catalog

Table 15 lists all 27 fault types injected during laboratory data collection.

## G. Extended Experimental Details

### G.1. Anomaly Detection: MLP Architecture and Training

The S-E-F-C MLP is a supervised regressor mapping 18 Setpoint signals (setpoint_pos_0...5, setpoint_vel_0...5, setpoint_acc_0...5) to 6 Effort signals (effort_motor_torque_0...5); the per-episode mean absolute error between predicted and true motor torque is the anomaly score. The network has three hidden layers (512, 256, 128 units) with ReLU activations and no dropout. It is trained on healthy episodes only (948 episodes from voraus-AD) with Adam (learning rate $5 \times 10^{-4}$, weight decay $1 \times 10^{-5}$, batch size 4,096), a cosine-annealing schedule, for up to 500 epochs with early stopping (patience 30).

### G.2. TCN-Transformer Architecture and Training

The TCN-Transformer comprises a 3-layer dilated Temporal Convolutional Network (kernel size 3) followed by a 2-layer Transformer encoder (4 attention heads, hidden dimension 64, feed-forward dimension 128), for a total of approximately 105,000 parameters. It is trained with AdamW (learning rate $1 \times 10^{-4}$) for 100 epochs to predict 1-step-ahead joint accelerations from a 10-step context window; predictions are integrated (Euler, $\Delta t = 0.01$ s) to recover position and velocity.

*Table 13.* S-E-F-C mapping for the Isaac Sim (UR5) synthetic source. Signals are procedurally generated and logged at a base 60 Hz simulation step before interpolation.

| Signal Group (Raw Prefix) | Described Fields | S-E-F-C Role | Unit |
|---|---|---|---|
| `episode`, `step`, `time` | Global/episode steps, sim/wall time | Metadata | count / s |
| `state_machine`, `phase` | Task, controller phases, event IDs | Context | enum / str |
| `joint_cmd_pos_rad_*` | Commanded joint positions (×6) | Setpoint | rad |
| `joint_cmd_vel_radps_*` | Commanded joint velocities (×6) | Setpoint | rad/s |
| `joint_pos_rad_*` | Actual joint positions (×6) | Feedback | rad |
| `joint_vel_radps_*` | Actual joint velocities (×6) | Feedback | rad/s |
| `joint_accel_radps2_*` | Actual joint accelerations (×6) | Feedback | rad/s$^2$ |
| `joint_torque_nm_*` | Joint applied torques (×6) | Effort | Nm |
| `joint_pos_error_rad_*` | Tracking error per joint (×6) | Context | rad |
| `ee_cmd_pos_*`, `quat_*` | Commanded TCP pose (pos, quat) | Setpoint | m / — |
| `ee_pos_*`, `quat_*` | Actual TCP pose (pos, quat) | Feedback | m / — |
| `ee_euler_*_rad` | Actual TCP orientation (RPY) | Feedback | rad |
| `ee_linvel_*`, `angvel_*` | TCP velocities (linear, angular) | Feedback | m/s / rad/s |
| `ee_linacc_*`, `angacc_*` | TCP accelerations (linear, angular) | Feedback | m/s$^2$ / rad/s$^2$ |
| `gripper_cmd_rad` | Commanded gripper position | Setpoint | rad |
| `gripper_pos_rad` | Actual gripper position | Feedback | rad |
| `gripper_attached` | Gripper logical contact state | Context | bool |
| `contact_force_*_n` | Physical EE contact force (XYZ, mag) | Effort | N |
| `contact_torque_*_nm` | Physical EE contact torque (XYZ, mag) | Effort | Nm |
| `cube_pos_*`, `quat_*` | Target object pose (pos, quat) | Context | m / — |
| `cube_linvel_*`, `angvel_*` | Target object velocities | Context | m/s / rad/s |
| `ee_cube_offset_*` | Relative distance to target | Context | m |
| `cube_mass_kg` | Domain rand: payload mass | Context | kg |
| `cube_friction_coeff` | Domain rand: target object friction | Context | — |
| `cube_width/depth/height_m` | Domain rand: target dimensions | Context | m |

*Table 14.* Per-category anomaly detection AUROC on voraus-AD. All unstructured baselines use 130 signals; S-E-F-C MLP uses 24. $N$ = number of anomalous episodes per category. [†]Bootstrap 95% CI. [‡]Std across categories.

| Fault Category | $N$ | 1-NN | PCA | GANF | CAE | LSTM-VAE | MVT-Flow | S-E-F-C MLP (Ours) |
|---|---|---|---|---|---|---|---|---|
| Additional friction | 144 | 74.8 | 76.4 | $88.5_{\pm4.5}$ | $89.4_{\pm0.2}$ | $88.7_{\pm1.5}$ | $96.6_{\pm0.6}$ | 80.0 |
| Miscommutation | 89 | 80.8 | 87.0 | $98.8_{\pm1.1}$ | $99.1_{\pm0.0}$ | $98.1_{\pm0.5}$ | $99.8_{\pm0.3}$ | **99.2** |
| Misgrip can | 11 | 100.0 | 100.0 | $47.6_{\pm13.1}$ | $100.0_{\pm0.0}$ | $100.0_{\pm0.0}$ | $95.3_{\pm3.3}$ | 83.8 |
| Losing can | 74 | 68.7 | 70.1 | $72.1_{\pm5.8}$ | $72.6_{\pm0.3}$ | $70.4_{\pm2.9}$ | $96.2_{\pm0.4}$ | 71.8 |
| Add. axis weight | 156 | 75.0 | 79.2 | $93.2_{\pm2.4}$ | $93.5_{\pm0.1}$ | $82.7_{\pm1.1}$ | $94.1_{\pm0.7}$ | **95.8** |
| Collision w/ foam | 72 | 69.6 | 73.9 | $81.2_{\pm6.2}$ | $81.5_{\pm0.2}$ | $81.5_{\pm2.1}$ | $87.5_{\pm1.2}$ | 74.7 |
| Collision w/ cables | 48 | 74.5 | 75.7 | $82.7_{\pm6.4}$ | $79.6_{\pm0.3}$ | $77.3_{\pm3.6}$ | $84.7_{\pm1.2}$ | 67.6 |
| Collision w/ cardboard | 22 | 82.8 | 83.6 | $77.5_{\pm5.8}$ | $78.6_{\pm0.3}$ | $82.8_{\pm4.8}$ | $88.3_{\pm1.2}$ | 76.7 |
| Varying can weight | 80 | 63.7 | 64.2 | $68.9_{\pm8.7}$ | $72.3_{\pm0.3}$ | $71.4_{\pm2.1}$ | $85.1_{\pm1.1}$ | 76.2 |
| Cable at robot | 10 | 63.0 | 71.6 | $76.6_{\pm8.1}$ | $83.1_{\pm0.3}$ | $96.0_{\pm1.4}$ | $100.0_{\pm0.0}$ | 91.3 |
| Invalid gripping pos. | 12 | 93.4 | 92.1 | $86.6_{\pm10.8}$ | $88.8_{\pm0.3}$ | $97.7_{\pm1.3}$ | $100.0_{\pm0.0}$ | 89.1 |
| Unstable platform | 37 | 83.6 | 85.9 | $84.6_{\pm3.8}$ | $83.9_{\pm0.2}$ | $93.6_{\pm1.9}$ | $96.1_{\pm0.7}$ | 92.8 |
| **Mean** | 755 | 77.5 | 80.0 | $79.9_{\pm12.7}$[‡] | $85.2_{\pm9.2}$[‡] | $86.7_{\pm10.1}$[‡] | $93.6_{\pm5.7}$[‡] | **83.2 [81.5, 86.1]**[†] |

*Table 15.* Complete Fault Catalog. Checkmarks (✓) indicate whether the fault is included in the Pick-and-Place (PP), Screwing (Scr), and Peg-in-Hole (PiH) task recordings. Crosses (×) indicate absence.

| Fault | Explanation | Injection Procedure | PP | Scr | PiH |
|---|---|---|---|---|---|
| Damaged screw thread | Screw thread physically damaged, preventing proper engagement during tightening. | Pre-damaged the screw thread with sandpaper. | × | ✓ | × |
| Missing screw | Tightening is attempted with no screw present. | Removed the screw from screwdriver before the cycle. | × | ✓ | × |
| Damaged plate thread | The threaded hole in the plate is damaged, preventing engagement. | Pre-damaged the plate hole with a metal screwdriver. | × | ✓ | × |
| Loosening phase | A counterclockwise loosening rotation replaces tightening; a normal phase with a distinct signal. Counted as an anomaly to match AURSAD schema. | Reversed the rotation direction in the controller program to execute a counterclockwise loosening pass. | × | ✓ | × |
| Gripper activation failure | The vacuum gripper fails to activate and never picks up the box. | Disabled the gripper activation command in the robot program. | ✓ | × | × |
| Gripper release during motion | The gripper releases the payload mid-trajectory, causing an abrupt payload loss. | Triggered a programmatic gripper-release command at a scripted timestep mid-trajectory. | ✓ | × | × |
| Additional axis payload | A dead weight attached to one link increases inertia and gravity loading on all joints. | Bolted calibrated weights of varying mass to one robot link. | ✓ | ✓ | × |
| Collision with foam object | Contact with a soft foam block produces a brief TCP force spike without a protective stop. | Placed a foam cube directly in the programmed TCP trajectory. | ✓ | ✓ | ✓ |
| Unexpected payload weight | The transported box has a weight that deviates from the nominal payload configuration. | Swapped the nominal box for a known heavier or lighter one. | ✓ | × | × |
| Invalid gripping position | Timing error: the gripper closes after the arm has begun lifting, gripping the object off-centre. | Added a sleep() call in the controller program so gripper closure lags the lift motion. | ✓ | × | × |
| Unstable mounting platform | Base instability introduces low-frequency vibrations into the arm. | Placed 8 layers of towels under the base plate. | ✓ | × | × |
| Joint position limit violation | A joint moves outside its configured soft or hard position limits, triggering a safety stop. | Random waypoint slightly beyond the configured soft joint limit. | ✓ | × | × |
| TCP frame misconfiguration | TCP frame or mounting orientation misconfigured at the controller; gravity torques deviate. | Entered an incorrect TCP offset or mounting angle in the controller settings. | ✓ | ✓ | ✓ |
| Payload weight misconfiguration | Payload mass misconfigured while a tool or workpiece is physically attached. | Left configured payload weight at multiple wrong mass configurations. | ✓ | ✓ | × |
| External arm disturbance | A continuous external force pulls or pushes the TCP during motion. | Anchored an elastic rope between the bench frame and the tool flange. | ✓ | ✓ | ✓ |
| Mild payload CoG misconfig. | Payload CoG specified at an incorrect offset from the tool flange; gravity compensation is wrong. | Entered a CoG offset in the payload configuration that differs from physical CoG. | ✓ | ✓ | ✓ |
| Collision with hanging cable | A loose cable drags along a robot link or the tool flange during motion. | Suspended a loose cable across the trajectory at arm height. | ✓ | ✓ | ✓ |
| Collision with cardboard | A cardboard carton in the trajectory provides moderate resistance; may trigger a protective stop. | Placed a free-standing or braced cardboard box in the programmed trajectory. | ✓ | ✓ | ✓ |
| Collision with rigid object | A rigid plastic block in the TCP path triggers an immediate protective stop. | Clamped a rigid plastic block in the programmed TCP path. | ✓ | ✓ | ✓ |
| Peg insertion misalignment | Peg approaches the hole at an angular or lateral offset and jams against the rim. | Added a recorded lateral offset at the insertion approach waypoint. | × | × | ✓ |
| Hole obstruction | Foreign object or debris in the hole prevents full insertion. | Placed a small object (metal screw) inside the hole before insertion. | × | × | ✓ |
| Incorrect insertion depth | Insertion terminates at an incorrect Z depth due to a misconfigured waypoint. | Shifted the insertion endpoint waypoint Z by recorded offset from nominal. | × | × | ✓ |
| Peg surface contamination | Contamination or roughness on the peg raises insertion friction and produces stick-slip. | Applied dry chalk powder to the peg surface. | × | × | ✓ |
| Fixture displacement | Insertion fixture shifted laterally, breaking the match between programmed approach and actual hole. | Physically shifted the hole fixture by recorded offset without updating program. | × | × | ✓ |
| Self-collision | Arm configuration causes a link to contact the robot body or mounting fixture. | Random waypoint that forces a self-colliding configuration. | ✓ | × | × |
| Missing box | Box absent from the pick position; the gripper closes on empty air. | Removed the box from the pick position before the episode started. | ✓ | × | × |
| Missing peg | Arm descends into the hole empty-handed; insertion produces no contact force. | Removed the peg from the gripper before the episode started. | × | × | ✓ |

### G.3. Forecasting Baselines

All trainable forecasting baselines use the same 1,093-episode training split and 10-step context window, minimizing MSE on predicted joint accelerations with Adam:

- **Linear:** a single `nn.Linear` layer mapping the flattened context window ($10 \times 36 = 360$ inputs) to the target acceleration space.

- **Flat MLP:** two hidden layers (128 and 64 units) with ReLU activations on the flattened context window.

- **TCN:** a 2-layer Temporal Convolutional Network (`Conv1d`, kernel size 3, hidden dimension 64).

- **Kinematic Baseline:** a non-learned predictor that assumes constant velocity (zero predicted acceleration).

## H. Isaac Sim Synthetic Pipeline Details

The synthetic Pick & Place episodes are generated in NVIDIA Isaac Sim with extensive domain randomization over the procedurally generated UR5 task:

- **Mass:** payload (cube) mass sampled uniformly per episode in $[0.10, 0.30]$ kg (broader exploratory runs up to $0.80$ kg); robot link masses fixed.

- **Friction:** target-object Coulomb friction sampled in $[0.30, 0.50]$; gripper-pad friction fixed at 1.2.

- **Controller gains:** gripper proportional gain $K_p$ sampled uniformly in $[5000, 12000]$; arm PID gains fixed.

- **Sensor noise:** Gaussian noise with base $\sigma_{\text{base}} = 0.002$ scaled per modality (joint positions $\sigma = 0.002$ rad, velocities $\sigma = 0.02$ rad/s, efforts $\sigma = 0.1$; object position $\sigma_{xy} = 0.002$ m, $\sigma_z = 0.001$ m).

- **Geometry/task:** cube dimensions and initial spawn position randomized per episode within an $8 \times 8$ cm box.

- **Timing:** the physics engine steps at 60 Hz and is interpolated to the 100 Hz laboratory standard during ingestion.

For sim-to-real validation we replay real `target_joint_*` waypoints in Isaac (UR3 USD asset, position control), pairing simulated and real episodes by ID; the resulting phase-wise gap is reported in Table 5.

