# OpenReview forum: "FactoryNet: A Large-Scale Dataset toward Industrial Time-Series Foundation Models"
_ICML.cc/2026/Workshop/FMSD — FMSD @ ICML 2026 Poster_

### Official Review · Reviewer_ua2D · 2026-05-19
**Useful dataset contribution, but the scope and framing should be tightened**

**Rating:** 7
**Confidence:** 4

**Review:**

### Summary

This paper introduces FactoryNet, an industrial time-series dataset with anomaly annotations collected across multiple machines, tasks, and data sources, together with a unified Setpoint-Effort-Feedback-Context (S-E-F-C) signal taxonomy. The paper addresses a relatively less explored data gap in industrial time-series learning, especially for actuated systems where separating commanded intent from measured response may be useful. The dataset itself appears potentially valuable, but the paper is primarily a dataset paper, and the current framing around industrial time-series foundation models is stronger than what is empirically validated.

### Strengths

1. The paper studies an interesting and less explored problem: the lack of open industrial time-series datasets with richer control-loop structure across multiple embodiments.

2. The proposed S-E-F-C schema seems reasonable. In particular, separating setpoints, effort, feedback, and context provides a more structured view of industrial signals than treating all channels as a single undifferentiated multivariate stream.

3. The dataset documentation is relatively comprehensive, including source mappings, anomaly categories, and data organization.

4. The paper includes several validation experiments, which is useful for illustrating potential uses of the dataset.

### Areas for Improvement

1. The scope of the paper is somewhat over-claimed. The dataset focuses on a specific segment of industrial time-series data related to actuated systems and cross-embodiment settings, but the framing sometimes reads as if it were a general industrial time-series foundation-model dataset.

2. The paper does not actually validate the central foundation-model motivation. There is no pretraining study, no zero-shot evaluation of a trained TSFM on this corpus, and no demonstration that existing TSFMs improve when exposed to this data.

3. The anomaly-detection validation is mixed. The proposed structured baseline is competitive with some weaker baselines, but it does not outperform the strongest full-signal methods.

4. A substantial synthetic component is included, but the sim-to-real gap remains a meaningful limitation.

### Detailed Comments

1. In my reading, the core contribution is the dataset plus the S-E-F-C signal organization, not a foundation model or a validated training recipe for one. I suggest making that distinction more explicit throughout the paper.

2. The S-E-F-C taxonomy is a sensible way to organize signals from actuated systems, and it may help future modeling work. However, the current paper does not yet show that this representation leads to better foundation-model pretraining or transfer at the model level.

3. The biggest concern is therefore about scope and framing. The paper collects and organizes data from a specific industrial setting, but that is still different from establishing a broadly useful foundation-model training corpus for industrial time series.

4. I also think the cross-embodiment claims should be stated more carefully, since the evidence is still relatively limited.

### Justification of Score

Overall, I think this is a useful dataset-oriented submission and likely relevant to the workshop. My main concern is that the paper’s broader foundation-model framing is not yet supported by the experiments. I would be more positive if the claims were narrowed to emphasize the dataset and schema contribution directly.

---

### Official Review · Reviewer_Uzr3 · 2026-05-20
**FactoryNet**

**Rating:** 6
**Confidence:** 4

**Review:**

# Summary
The authors present their work, FactoryNet, a dataset aiming to serve as a basis for industrial time-series foundation models. A novel taxonomy known as S-E-F-C is provided that allows for mapping data columns into the specific embodiment categories, which allows for a single data loader to work with many different types of machinery. The authors provide experiments for an MLP architecture as well as a TCN-Transformer.

# Strengths
+ The S-E-F-C taxonomy is novel and provides a useful layer of abstraction.
+ The dataset itself seems incredibly useful for industrial settings.

# Areas for Improvement
- Would have been nice to see some SSL-based pretraining experiments
- The number of tasks is limited to three (pick and place, screwdriving and peg-in-hole).
- I see no evidence that this dataset was used to train/fine-tune at least one foundation model.

# Detailed Comments
Overall I think this is a very useful and practical dataset. I think this paper would be strengthened by a few things, namely the addition of foundation model experiments, as well as other metrics besides MAE or AUC-ROC.

# Justification of score
While the authors provide a useful SEFC taxonomy and a useful industrial dataset, I am concerned about the lack of experiments on foundation models. Moreover, the evidence provided shows that the TCN-Transformer under-performs more primitive architectures such as MLPs. I'm less concerned about the lack of sim-to-real benchmarks and more so the lack of experiments on modern foundation models to show its plausibility.

---

### Official Review · Reviewer_Rnad · 2026-05-20
**FactoryNet: Strong Physics-Informed Schema with Room to Expand Cross-Embodiment Evaluation**

**Rating:** 7
**Confidence:** 4

**Review:**

The paper introduces FactoryNet, a large-scale, multi-machine dataset containing 51 million datapoints across 23,000 tasks. It is designed to act as a universal pretraining corpus for industrial time-series foundation models.

The core innovation is the S-E-F-C (Setpoint, Effort, Feedback, Context) schema. This taxonomy acts as a physics-informed "universal vocabulary" that maps hundreds of vendor-specific data columns into distinct structural roles (commanded intent, energy expended, physical response, and environmental factors). This removes chaotic background noise and enables models to cleanly capture the underlying control loop.

To validate the dataset, the authors trained multiple baseline architectures. This includes a simple supervised MLP that predicts expected motor torque (Effort) from commanded paths (Setpoint), outperforming complex, traditional anomaly detection baselines.

Core Strengths -
1/ Physics-Informed Standardization: The S-E-F-C schema successfully moves beyond flat statistical correlations, encoding a structural causal bias (cause, energy, and effect) directly into the model's inputs.

2/ Zero-Shot Transfer Capabilities: By standardizing data columns into machine-agnostic functional categories, the dataset enables cross-embodiment dynamics transfer, allowing a model trained on one machine to generalize to an entirely different mechanical configuration.

3/ High Empirical Scale: It combines real-world industrial tasks with high-fidelity simulations to build an extensive, unified time-series substrate.

Key Areas for Improvement & Technical Open Questions
1/ Evaluation Scope: The zero-shot cross-embodiment transfer challenge is currently restricted to a single evaluation pair (transferring from a Yu-Cobot to a UR3e robot). To truly claim a "universal" foundation corpus, evaluations need to scale across other diverse machines in the dataset.

2/ Statistical Rigor: The cross-embodiment benchmarks rely on a single random seed initialization. Running these across multiple seeds to report standard deviations is necessary to confirm the stability of the transfer metrics.

3/ Handling Missing Modalities: Certain machinery platforms in the dataset do not expose all critical control-loop channels. A deeper analysis is required to understand how missing feedback or effort channels affect the overall robustness of the foundation model.